

# Computational identification of miRNAs that modulate the differentiation of mesenchymal stem cells to osteoblasts

Kanokwan Seenprachawong[1], Pornlada Nuchnoi[2], Chanin Nantasenamat[3], Virapong Prachayasittikul[4] and Aungkura Supokawej[1]

[1] Department of Clinical Microscopy, Faculty of Medical Technology, Mahidol University, Bangkok, Thailand

[2] Center for Research and Innovation, Faculty of Medical Technology, Mahidol University, Bangkok, Thailand

[3] Center of Data Mining and Biomedical Informatics, Faculty of Medical Technology, Mahidol University, Bangkok, Thailand

[4] Department of Clinical Microbiology and Applied Technology, Faculty of Medical Technology, Mahidol University, Bangkok, Thailand

Corresponding author
Aungkura Supokawej,
aungkura.jer@mahidol.ac.th

## ABSTRACT

MicroRNAs (miRNAs) are small endogenous noncoding RNAs that play an instrumental role in post-transcriptional modulation of gene expression. Genes related to osteogenesis (i.e., *RUNX2*, *COL1A1* and *OSX*) is important in controlling the differentiation of mesenchymal stem cells (MSCs) to bone tissues. The regulated expression level of miRNAs is critically important for the differentiation of MSCs to preosteoblasts. The understanding of miRNA regulation in osteogenesis could be applied for future applications in bone defects. Therefore, this study aims to shed light on the mechanistic pathway underlying osteogenesis by predicting miRNAs that may modulate this pathway. This study investigates RUNX2, which is a major transcription factor for osteogenesis that drives MSCs into preosteoblasts. Three different prediction tools were employed for identifying miRNAs related to osteogenesis using the 3'UTR of *RUNX2* as the target gene. Of the 1,023 miRNAs, 70 miRNAs were found by at least two of the tools. Candidate miRNAs were then selected based on their free energy values, followed by assessing the probability of target accessibility. The results showed that miRNAs 23b, 23a, 30b, 143, 203, 217, and 221 could regulate the *RUNX2* gene during the differentiation of MSCs to preosteoblasts.

## INTRODUCTION

Osteogenesis is a complex multistep process that includes proliferation, maturation and matrix mineralization from the development of mesenchymal stem cells into bone tissue. Mesenchymal stem cells (MSCs) are a type of adult stem cell that can be isolated from bone marrow and various tissues, such as muscle, adipose tissue, placenta and umbilical cord (*Erices, Conget & Minguell, 2000*). MSCs are multipotent stem cells that have the ability to self-renew or differentiate into mesodermal-derived cells, such as osteoblasts, chondrocytes and adipocytes (*Deans & Moseley, 2000; Pittenger et al., 1999*). MSCs have

high potential for use in cell therapy due to their special properties: proinflammatory, immunoprivilege, and multi-differentiation. Most importantly, the use of pluripotent stem cells derived from embryonic stem cells (ESCs) or induced pluripotent stem cells (iPSCs) is still limited because of ethical issues, technical problems and teratoma formation of these cells. Transplantation of these MSCs has clear advantages for the future treatment of bone defects, bone fractures, osteoporosis and osteoarthritis according to both *in vitro* studies and clinical trials. The potential of treatment showed a low outcome due to many unknown mechanisms of MSCs, particularly the osteogenic regulatory system of MSCs. There are many signaling pathways, such as the Wnt signaling pathway and BMP pathway, that play an integrative role for bone development (*James, 2013*). These signaling pathways ultimately affect major transcription factors, such as runt-related transcription factor2 (RUNX2) and osterix (OSX) (*Komori, 2006*). RUNX2 is a major transcription factor that regulates the differentiation of MSCs to preosteoblasts, and osterix plays a significant role in the development of the preosteoblast stage into osteoblasts. In this work, the regulatory system of osteogenesis is extensively discussed, including not only the signaling pathway but also epigenetic control, such as DNA methylation, histone modification and miRNAs.

MicroRNAs (miRNAs) are small endogenous non-coding RNAs, and their length is approximately 21–24 nucleotides. MiRNAs regulate gene expression at the post-transcription level through the degradation of mRNA or inhibition of protein synthesis (*He & Hannon, 2004*). Their function is through specific binding of miRNA and the 3' UTR of the target gene. MiRNAs are associated with stem cell differentiation and tissue development, including bone development. The regulation of miRNAs in osteogenesis has been studied, particularly in the expression of *RUNX2*. In the studies of *Huang et al. (2010)* and *Tome et al. (2011)*, miR-204 and miR-335 exhibited an inhibitory mechanism through binding at the 3' untranslated region of *RUNX2*. In addition, miR-103a inhibited bone formation by binding the *RUNX2* target under both physiological and pathological mechanical conditions during *in vitro* and *in vivo* studies (*Zuo et al., 2015*). *Zhang et al. (2011)* found that the osterix gene was regulated by miR-637. MiR-637 enhanced adipogenesis and inhibited osteogenesis.

RUNX2 is a master transcription factor that controls osteogenesis. *RUNX2* or *CBFA-1* knockout mice showed a complete defect of bone formation because of osteoblast maturational arrest (*Komori et al., 1997*). The activation of RUNX2 in osteogenesis is regulated by several signaling pathways (i.e., Wnt and bone morphogenic protein) (*Hayrapetyan, Jansen & Van den Beucken, 2015*). The epigenetic regulation of osteogenesis has been widely discussed but it is not well characterized, particularly, the mechanism of miRNAs. Recently, *Kang & Hata (2015)* proposed that the major mechanism of the regulatory function of miRNAs can be attributed to its controlling of the osteogenesis process via the cell fate determination of stem cells. Moreover, the functions of miRNAs are complex and remain unclear; thus, more studies on the role of miRNAs in osteogenesis are needed for future applications in clinical trials and diagnoses because previous studies cannot clearly describe the multiple steps of osteogenesis. Microarrays and direct cloning are typically used for predicting miRNAs, but these approaches are time consuming and expensive. Therefore, the objective of this study is to apply bioinformatics tools

for predicting the miRNAs involved in osteogenesis, which is performed using the 3' untranslated region (3'UTR) of *RUNX2* gene and miRNA database.

## MATERIALS AND METHODS

### Data collection

The workflow implemented for miRNA prediction is shown in Fig. 1. The human 3'UTR sequence of *RUNX2* was obtained from the NCBI database (www.ncbi.nlm.nih.gov). Using the nucleotide database and keywords including homo sapiens, *RUNX2*, and mRNA for searching nucleotide sequences, the results showed 122 nucleotide sequences. Homo sapiens runt-related transcription factor 2 (*RUNX2*), transcript variant 1, mRNA (accession number NM_001024630) was selected. It is a 5,553 bp linear mRNA. The structure of human protein coding mRNA includes a 5'cap, 5'UTR, coding sequence, 3'UTR, and a poly-A tail. The 3'UTR sequence of the *RUNX2* gene is located downstream from the coding sequence region and is composed of 3,777 nucleotide bases (Table S1).

### Prediction of miRNAs

The prediction of miRNAs was investigated using 3 different algorithms that are the most widely used in the updated version as follows: miRanda, RegRNA and TargetScan.

The miRanda software (*Betel et al., 2010*) has a miRNA prediction function that uses an algorithm called mirSVR. The mirSVR algorithm learns to predict mRNA target sites on mRNA expression changes from a panel of mRNA transfection experiments and displays the scores and ranks the efficiency of the miRanda-predicted miRNAs. mirSVR used 3 main features: (1) duplex structure of the target site and miRNA at the seed region, (2) composition flanking the target site, and (3) secondary structure accessibility of the site and conservation to calculate and display the mirSVR score. To predict miRNAs, the name of the target mRNA and species were used as the input. The lower mirSVR scores are correlated with down-regulation at the mRNA or protein levels and can be interpreted as a probability of target inhibition, leading to candidate miRNA selection.

RegRNA version 2.0 (*Huang et al., 2006*; *Chang et al., 2013*) is an integrated web server for miRNA prediction. This software database was retrieved from a literature survey of experimentally validated miRNAs, namely, miRBase. The miRBase database provides extensive miRNA sequence data, annotations and predicted gene targets. To predict miRNAs, the mRNA target sequence was used as the input. The prediction results are presented via both textual and graphical interfaces. The minimum free energy (MFE) of the miRNA-target site duplex and score are determined by miRanda and RegRNA integrated tools. The known miRNA genes in three mammalian genomes, including human, mouse and rat, were obtained from miRBase. Therefore, the RegRNA currently has 21,643 known miRNA sequences. During the miRNA prediction, the lower MFE values reveal the energetically more probable hybridizations between the miRNAs and target genes.

TargetScanHuman version 6.2 (*Lewis, Burge & Bartel, 2005*; *Grimson et al., 2007*) is a web server for predicting miRNAs by searching for the presence of conserved sites that match the seed region of each miRNA. In mammals, predictions are ranked based on the

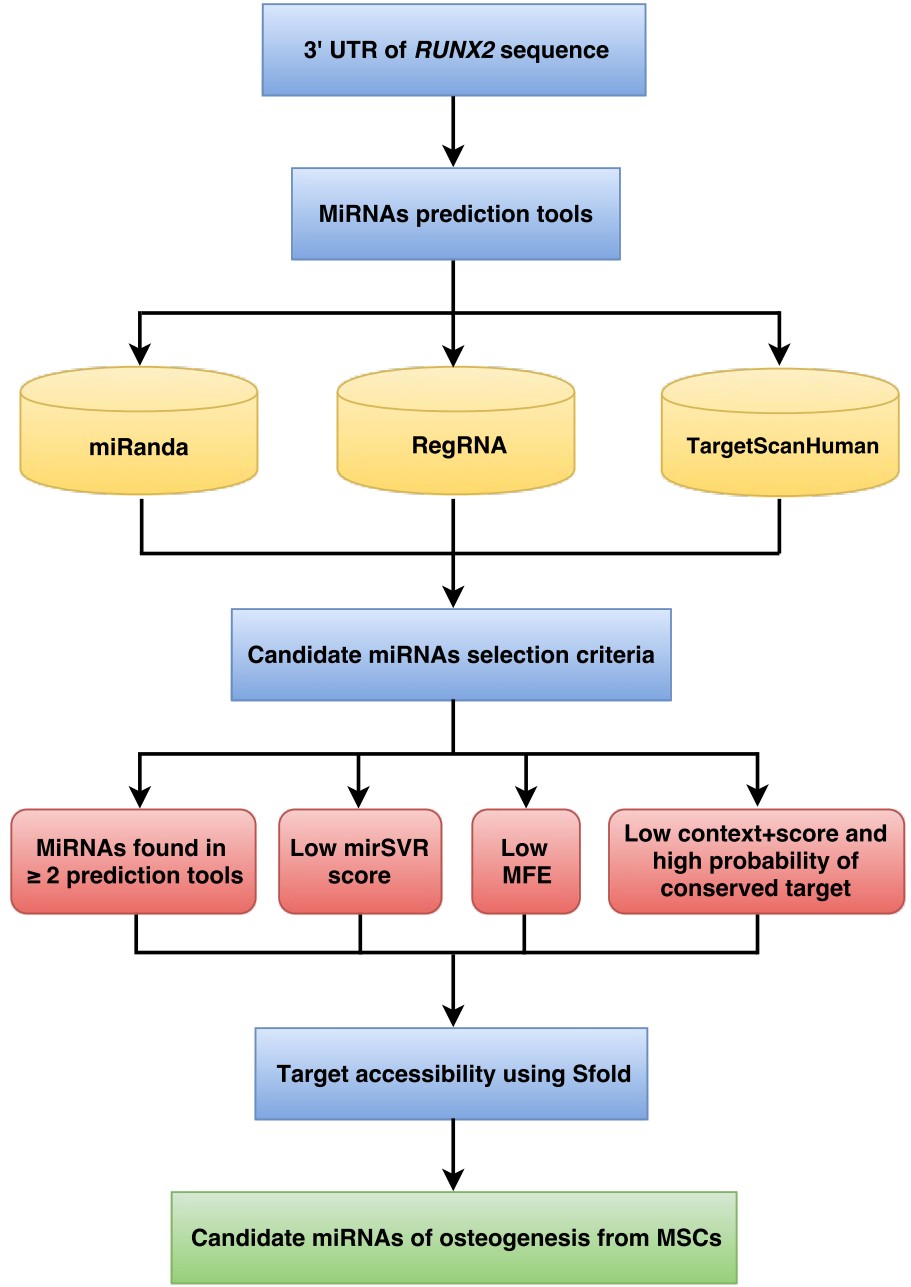

**Figure 1** Schematic representation of the workflow for the identification of miRNAs involved in osteogenesis.

predicted efficacy of targeting as calculated using the context + scores of the sites and their probability of conserved targeting. TargetScanHuman considers matches to annotate human UTRs and their orthologs, as defined by UCSC whole-genome alignments. To predict miRNAs, the name of the target mRNA and species were used as the input. The
lower context + score and higher probability of conserved target reveal more probable candidate miRNAs.

## MiRNA selection criteria

The predicted miRNAs from the previous step followed by the selection criteria can provide candidate miRNAs in the next step. The candidate miRNA selection criteria are as follows: (1) exhibited greater than or equal to 2 in 3 prediction tools; (2) high negative free energy that represented more probable hybridizations of miRNA-mRNA duplex; (3) high negative mirSVR score showed a high probability of target inhibition; and (4) high negative context + score and high probability of conserved target revealed good candidate miRNAs for target gene inhibition.

## Target accessibility

The miRNAs that were qualified by the candidate miRNA selection criteria were assessed in terms of target accessibility using Sfold. The Sfold software is a statistical sampling algorithm for predicting the RNA secondary structure that is accessible for RNA-targeting nucleic acids through base-pairing interactions. Target accessibility was predicted by probability; sites with high probability are estimated to be a good accessible region for miRNA binding (*Ding & Lawrence, 2001*; *Ding & Lawrence, 2003*; *Ding, Chan & Lawrence, 2004*). The miRNAs that can hybridize to target regions of mRNA were available as candidates for target gene inhibition.

## RESULTS

### Prediction of *RUNX2*-specific miRNAs

Following the implementation of the workflow for miRNA prediction that was explained in the materials and methods (Fig. 1), to identify the predicted miRNAs of the *RUNX2* gene, we computationally identified miRNAs using three prediction tools, namely, miRanda, RegRNA and TargetScanHuman. The 3'UTR sequence of *RUNX2* was retrieved from the NCBI database, which is 3,777 nucleotides in length. Then, miRNAs were predicted using miRNA prediction tools. The results indicated that 922, 71 and 30 miRNAs were predicted by RegRNA, miRanda and TargetScanHuman, respectively (Table S2).

### Selection of *RUNX2* miRNAs

The predicted miRNAs will be manually selected for further candidate miRNAs. Based on the selection criteria that were explained in the materials and methods, we retrieved only 27 miRNAs from all three prediction tools that satisfied all criteria. The selected miRNAs that resulted from at least two prediction tools and that matched all remaining criteria were used. The results showed 40 miRNAs from overlapping between RegRNA and miRanda, 2 miRNAs from overlapping between RegRNA and TargetScan, and 1 miRNA from overlapping between miRanda and TargetScan. The list of these selected miRNAs and their respective MFE values are presented in Fig. 2A. Then, we categorized 70 miRNAs in two tables: those that resulted from all three prediction tools (Table S3) and those that resulted from 2 of 3 prediction tools (Table S4), which are sorted by their high negative

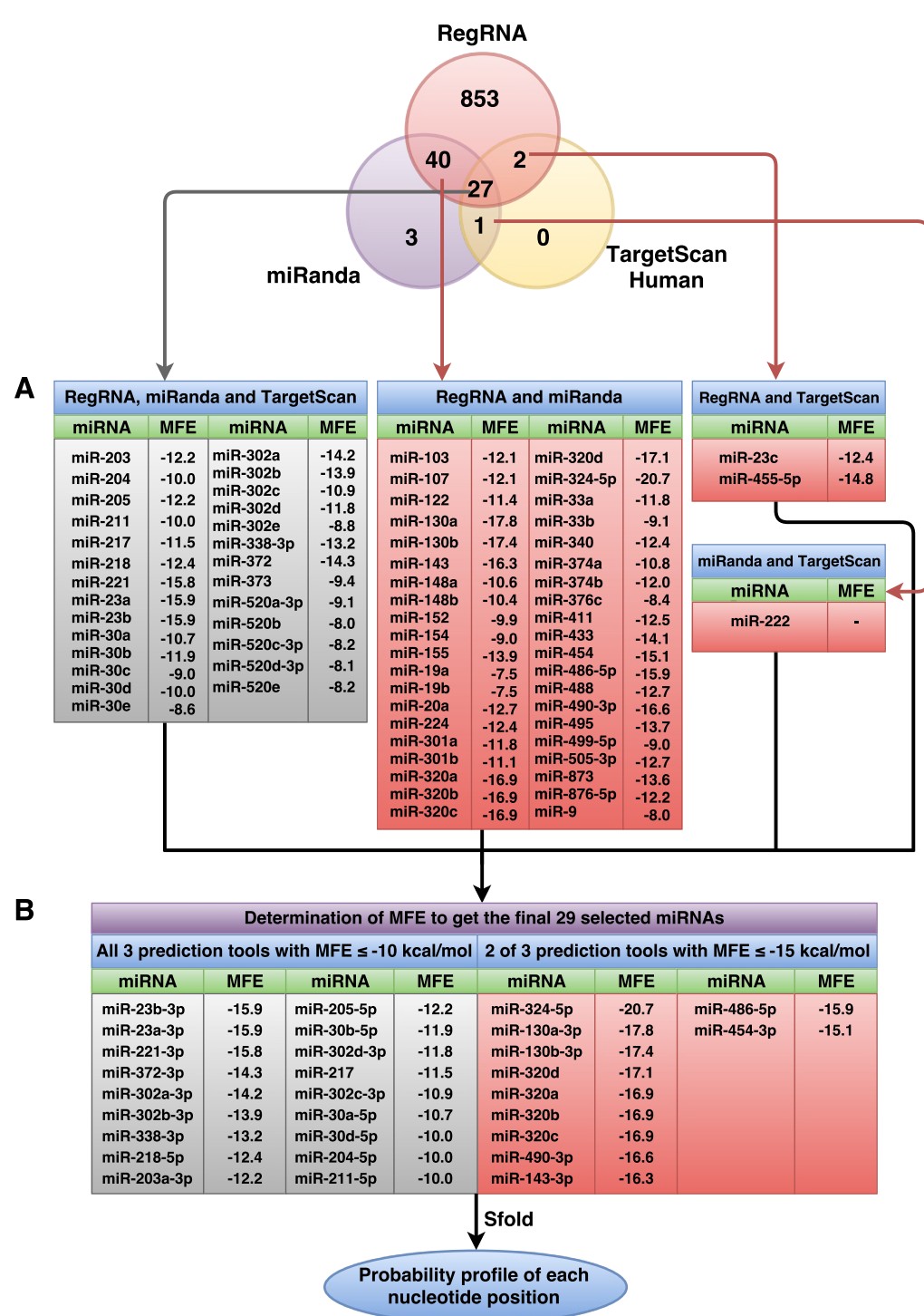

**Figure 2** Overall number of *RUNX2*-specific miRNAs using RegRNA, miRanda, and TargetScanHuman and the respective number of miRNAs obtained from the prediction tools. All *RUNX2*-specific miRNAs with their respective MFE values are shown in (A). MiRNAs were further filtered as to derive the final set of 29 selected miRNAs (B) using MFE thresholds of $\leq -10$ (all three prediction tools) and $\leq -15$ (two of three prediction tools) kcal/mol.

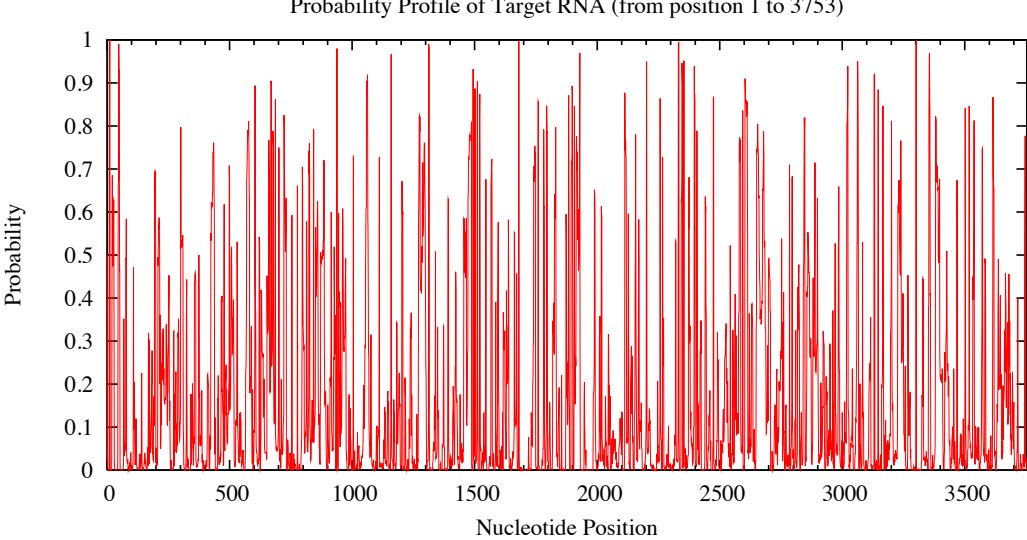

**Figure 3** **Probability profile of 3'UTR *RUNX2* target RNA.** The target regions are indicated on the histogram. Target structure features are essential for target binding by miRNA. The higher the probability, the stronger the miRNA binding to the secondary structure of the target region.

free energies. Minimal free energy was further used for the determination of candidate miRNAs based on the minimal free energy: $\leq -10$ and $\leq -15$ kcal/mol for miRNAs that were predicted using all three prediction tools and those that were predicted using two prediction tools, respectively. The list of 29 selected miRNAs and their respective values (Fig. 2B) were further analyzed using the Sfold software.

### *RUNX2*-specific miRNA candidates

Candidate miRNAs from the prediction were evaluated based on the probability of target accessibility using the Sfold software. The analysis of the predicted target structural accessibility also revealed cases of specific miRNA binding that is important for target gene inhibition. A high probability will increase the chance of successful miRNA binding. We input the 3'UTR of the *RUNX2* sequence, and then we obtained a histogram of the probability profile in each nucleotide position (Fig. 3). The positions of nucleotides with a probability greater than 0.5 were designated to be an accessible region for miRNA binding. Consequently, miRNAs 23b, 23a, 30b, 143, 203, 217, and 221 were selected as potential *RUNX2*-specific miRNA candidates for target gene inhibition (Fig. 4).

## DISCUSSION

RUNX2 is a transcription factor or master switch that controls osteogenesis, including MSC condensation, osteoblast proliferation and differentiation, and osteoblast maturation (*Komori, 2002*). There are multiple mechanisms that control *RUNX2* activity, including transcriptional regulation, translational modifications, alternative splicing, subnuclear localization, and interactions with cofactors, among others (*Stein et al., 2004*). *RUNX2* is specifically expressed in the early stage of osteoblast differentiation (*Pratap et al., 2003*). Overexpression of *RUNX2* or *Cbfa-1* (core binding factor α1) in

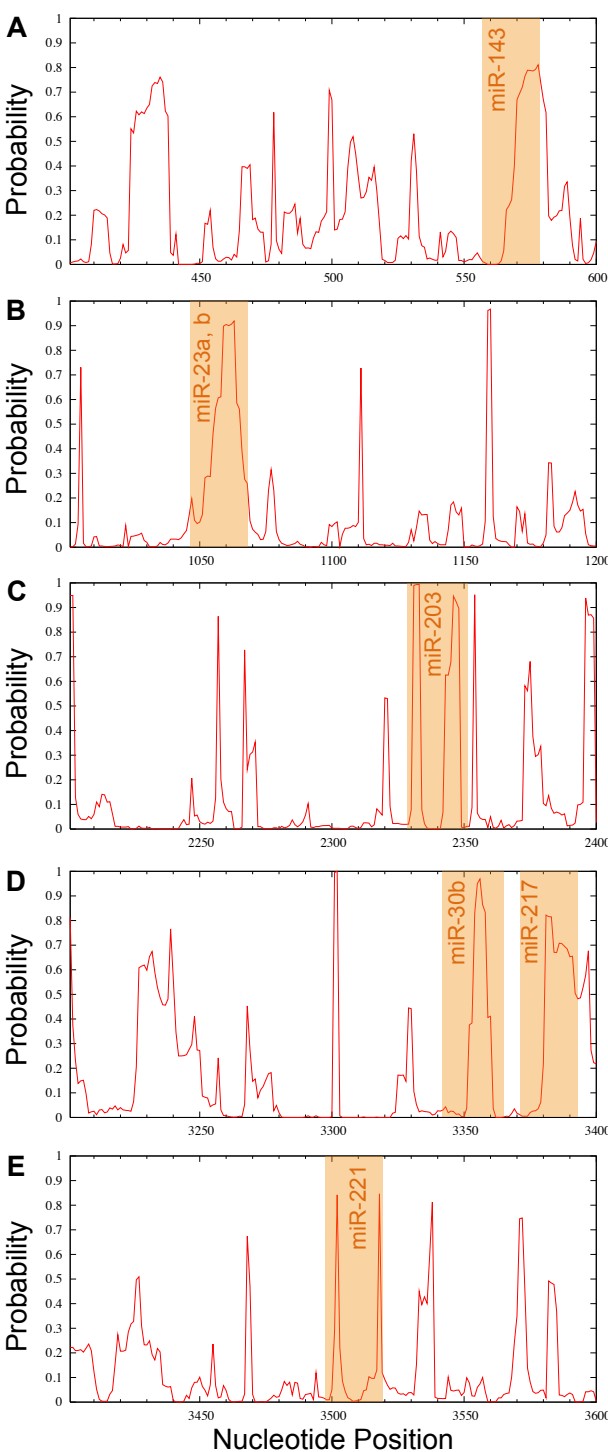

**Figure 4** **Probability profile of the miRNA binding region of the *RUNX2* target RNA.** Binding regions are shown by the orange shaded regions for miR-143 (A), miR-23a, b (B), miR-203 (C), miR-30b and miR-217 (D) as well as miR-221 (E) at positions 558–578, 1044–1068, 2329–2352, 3343–3366 and 3371–3393, as well as 3497–3519, respectively. A probability over 0.5 in the complementary seed sequence was denoted as the accessible region for miRNA binding.

MC3T3-E1 preosteoblastic cells showed increased expression of bone marker genes, including collagen type 1, osteopontin, and bone sialoprotein (*Ducy et al., 1997*). A recent study showed that *RUNX2* or *Cbfa-1* null mice can induced a complete lack of bone formation because of osteoblast maturational arrest, indicating a role of *RUNX2* as an osteoblast-specific transcription factor (*Komori et al., 1997*). The process of osteogenesis occurring depends on the expression of key transcriptional factors and changes in the epigenetic mechanisms. Epigenetic regulation of osteogenesis consists of DNA methylation, histone modification, and miRNA regulation, and they can influence gene expression by turning on or off the specific regulatory genes. *Villagra et al. (2002)* have provided evidence that the expression of the bone-specific rat osteocalcin gene can be regulated via DNA methylation.

MiRNAs have an important role in several cellular processes, such as development, cell proliferation, and cell death (*Friedman & Jones, 2009*). The expression of miRNAs is associated with several health problems, including osteoporosis (*Van Wijnen et al., 2013*). Therefore, studying a functional miRNA to regulate gene expression in bone development is still challenging for use in diagnostic and therapeutic approaches. MiRNAs showed a regulatory role in the osteogenesis of MSCs by regulating various biological processes, including inhibition of protein translation and promotion of mRNA degradation. Several studies have used microarrays to characterize the genes and expression profiles of miRNAs involved in osteogenesis.These studies identified the miRNA expression profile and their targeted gene in differentiated and undifferentiated human MSCs. They reported that miR-335 is downregulated in human MSC differentiation of adipogenesis and osteogenesis (*Tome et al., 2011*). *Huang et al. (2012)* applied microarrays to analyze miRNA expression profiles during adipogenic and osteogenic differentiation from human adipose tissue-derived MSCs. The results from the microarray analysis revealed that miR-22 decreased in adipogenesis but increased in osteogenesis, thus indicating a positive role of miR-22 in the regulation of osteogenesis. Microarrays can be used to identify the expression levels of thousands genes with high sensitivity and specificity, but it has limitations with respect to expense and difficulty in data interpretation. Thus, bioinformatics tools are powerful tools for investigating of genes and miRNA profiles. Effectively predicting miRNA-mRNA interactions remains challenging due to the complex process and limited knowledge of the interactions. Therefore, using bioinformatics tools is necessary for predicting miRNAs to find possible miRNA-mRNA interactions.

A variety of miRNA prediction algorithms are available, as well as different approaches. In this study, we used online bioinformatics tools, namely, RegRNA, miRanda, and TargetScan, to predict miRNAs that mediate posttranscriptional control of *RUNX2* expression involved in the process of osteogenesis. RegRNA is widely used for the prediction of functional RNA motifs because the RegRNA database is always updated, particularly the identification tool. This software also has a user-friendly interface, is easy to use, and provides good graphical visualization. miRanda is an algorithm for finding genomic targets of miRNAs by presenting mirSVR scores, which are calculated from 3 factors: binding energy of miRNA-miRNA interactions, conservation of the entire target site and the 3'UTR region. This algorithm is beneficial for predicting imperfect binding

within the seed region, but it has low precision and many false positives as estimated by experimental results (*Alexiou et al., 2009*; *John et al., 2004*). TargetScan is frequently used for miRNA prediction because of its convenience. This algorithm predicts miRNAs complementary with conserved sites of 3'UTR regions. The search result is limited only to the sites that are perfectly complementary in the miRNA seed sequence and extended to 22 nucleotide-long sequences that represent true interactions. In addition, there are many parameters for determining the outcome score, including seed match, complementary outside the seed sequence and positioning contribution; however, sites with poor seed pairing are missed (*Lewis, Burge & Bartel, 2005*). We selected three prediction tools because the combination of three prediction tools would be helpful in decreasing false positives and false negatives and provide high accuracy and precision for the selection of specific miRNAs.

All miRNAs predicted to target the 3'UTR of *RUNX2* mRNA are 23b, 23a, 30b, 143, 203, 217, and 221, and these might be potential miRNAs in osteogenesis from MSCs because these miRNAs have high negative free energies, thus representing a high probability of miRNA and mRNA interactions. Moreover, the positions of the target region that miRNAs can hybridize have a probability greater than 0.5, indicating that these miRNAs potentially play a regulatory role in osteogenesis. Interestingly, our candidate miRNAs have previously been shown to have some association with osteogenesis. The expression of the miR-30 family was examined during osteogenesis using miRNA PCR arrays. They found that the miR-30 family could repress *RUNX2* mRNA by immediate reduction and rapid recovery during osteogenic differentiation (*Eguchi et al., 2013*). MiR-221 was investigated using a miRNA microarray, which regulated osteogenic lineage commitment. Down-regulation of miR-221 using anti-miR-221 in hMSCs showed increased osteogenic marker genes but no significant change in the translation of osteopontin and osteocalcin. The results of this study revealed that miR-221 could play an important role in osteogenesis (*Bakhshandeh et al., 2012b*; *Bakhshandeh et al., 2012a*). In addition, miR-204, miR-211 and miR-338-3p, which were predicted in the 3 software packages, were correlated in a previous study. They found that miR-204 and miR-211 were negative regulators of *RUNX2*, which inhibit osteogenesis and promote adipogenesis of mesenchymal stem cells (*Huang et al., 2010*). Furthermore, the role of miR-338-3p in human ovarian epithelial carcinoma (EOC) was reported to inhibit ovarian cancer cell growth by targeting *RUNX2* (*Wen et al., 2015*). However, miR-204, miR-221 and miR-338-3p are not candidate miRNAs in this study because they were excluded because of their low probability of target accessibility. A plausible explanation could be that no single miRNA are deemed to be the sole driver for activating a gene but that the orchestrated work amongst several miRNAs is crucial for modulating the target gene. Particularly, *Wu et al. (2010)* found that 28 of 266 miRNAs inhibited the expression of *p21Cip1/Waf1*, which is a master downstream effector of tumor suppressors. In addition, *Jiang, Feng & Mo (2009)* employed a resemble method in which luciferase reporter assay was used to examine a number of miRNAs that inhibit the proto-oncogene, *cyclin D1*. They found that seven significant miRNAs could suppress the 3'UTR activity of *cyclin D1*. Moreover, several factors such as Watson-Crick base pairing, energy and seed region of miRNA, are essential for controlling the *in vivo* interaction of miRNA and mRNA. In summary, in order to achieve a high confidence and accuracy set of candidate

miRNAs which can affect the target gene, the strict selection criteria employed herein should be applied, especially for further experimental validation. However, the criterion could be made more or less stringent by the researcher depending on the circumstances or purpose of an investigation of interest, which would directly determine the breadth of miRNAs identified from the investigation.

Thus, the workflow for identifying miRNAs using bioinformatics tools is an alternative approach that is highly useful for the prediction of novel miRNAs in osteogenesis rather than using microarrays. This technique is convenient and inexpensive. For future studies, gene-targeting miRNAs and miRNA expression *in vitro* and *in vivo* are highly needed to evaluate the accuracy and precision of the identification of miRNAs.

## CONCLUSION

Osteogenesis is a very important process in the human body, which encompasses several cellular processes. The differentiation of MSCs to preosteoblasts is considered to be a critical step in osteogenesis. This step is controlled by many regulatory mechanisms, i.e., transcription factors, signaling pathways, and epigenetic mechanisms. MiRNAs showed a regulatory role in osteoblastic differentiation by regulating gene expression during the post-transcriptional process. A bioinformatics tool is a good choice for predicting candidate miRNAs involved in osteogenesis. Using three miRNA prediction tools in combination with selection criteria, we selected *RUNX2* as a target because the RUNX2 transcription factor acts as a master switch controlling osteogenesis. We retrieved seven candidate miRNAs, including miR-23b, 23a, 30b, 143, 203, 217, and 221. The knowledge obtained from this study can provide basic information regarding miRNAs in osteogenesis; however, further studies are needed to evaluate miRNAs and target genes *in vitro* and *in vivo* for clarifying the complex mechanism of osteogenesis.

## ACKNOWLEDGEMENTS

This research project is supported by Mahidol University.

### Funding

This research project is supported by Mahidol University. The funders had no role in study design, data collection and analysis, decision to publish, or preparation of the manuscript.

### Grant Disclosures

The following grant information was disclosed by the authors:
Mahidol University.

### Competing Interests

The authors declare there are no competing interests.
## Author Contributions

- Kanokwan Seenprachawong conceived and designed the experiments, performed the experiments, analyzed the data, wrote the paper, prepared figures and/or tables.
- Pornlada Nuchnoi conceived and designed the experiments.
- Chanin Nantasenamat analyzed the data, wrote the paper, reviewed drafts of the paper.
- Virapong Prachayasittikul contributed reagents/materials/analysis tools.
- Aungkura Supokawej conceived and designed the experiments, analyzed the data, wrote the paper, reviewed drafts of the paper.

## Data Availability

The research in this article did not generate any raw data.

## Supplemental Information

Supplemental information for this article can be found online at http://dx.doi.org/10.7717/peerj.1976#supplemental-information.

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
