# Peer review of "Computational identification of miRNAs that modulate the differentiation of mesenchymal stem cells to osteoblasts"

_PeerJ, doi:10.7717/peerj.1976_

## Round 0.1 · original submission · Minor Revisions

· Academic Editor

Minor Revisions

Due to late external reviewers answers, we decided to not delay the evaluation of your ms and to base the decision on the assessment of one reviewer and myself, as I am sufficiently knowledgeable in the field.

Please address all the points raised, especially comments on validity of the findings. Include in this task also miR 338-3p (Wen C, Liu X, Ma H, Zhang W, Li H. Int J Oncol. 2015 May;46(5):2277-85). You should also comment on possible reasons why some experimentally observed miR targeting Runx2, e.g. miR-375 (Du et al, Exp Ther Med. 2015 Jul;10(1):207-212) , were not identified.

·

Basic reporting

1. While the overall manuscript is well written and clear, I would recommend the authors to consider re-writing the following two sentences:
a. Abstract: The first sentence of the abstract seems to imply microRNAs are only regulators of osteogenesis-related genes. It would be more correct to say: MicroRNAs (miRNAs) are small endogenous noncoding RNAs that play an instrumental role in post-transcriptional modulation of gene expression, including that of genes related to osteogenesis (i.e. RUNX2, COL1A1 and OSX).
b. Materials and Methods, Data Collection: The last sentence of this paragraph is difficult to understand. While I agree that the 3’UTR of the RUNX2 is 3777nts long, I do not understand what the authors mean when they say “The 3’UTR sequence of the RUNX2 gene is later the coding sequence region …”.
2. There is a significant lack of appropriate references in particular in the introduction, i.e. not a single reference is being provided in the first 1.5 paragraphs, and lack of references in the first couple of sentences of the discussion.
3. In my opinion, the authors should expand figure 2 to also include the additional selection criterion of minimal free energy. For completeness, not only the 70 initial candidates, but also the final 29 candidates that were then further evaluated using Sfold software should be listed in this figure and therefore presented to the reader in the main manuscript text.

Experimental design

The overall experimental design is sufficiently described.
However, the rationale for selecting RUNX2 as the target gene of interest in this study should be provided earlier than in the Conclusions, e.g. in the introduction or methods section.

Validity of the findings

This study provides a relevant strategy for combing several easily accessible prediction tools resulting in a robust and stringent selection process for the identification of relevant microRNA-mRNA interactions.
However, I feel in the discussion the authors should address in more detail the fact that miR-204 and miR-211 were identified, but filtered out by their approach although both have already been described in in vitro studies to be targeting RUNX2. What implications does this have for the validity of the proposed selection strategy? Did these two microRNAs for example only just not make the final cut-off of MFE and target accessibility?
Although the authors do acknowledge that in vitro and in vivo studies are required to further evaluate the accuracy of microRNA identification using prediction tools, I feel this point should be discussed in more detail on the example of miR-204/211.

---

## Round 0.2 · accepted · Accept

· Academic Editor

Accept

You addressed appropriately the concerns that had been raised by the reviewer and the Editor. I thank you for submitting your work to PeerJ.